

# Permanent ectoplasmic structures in deep-sea *Cibicides/oides* taxa – long-term observations at in situ pressure

Jutta E. Wollenburg[1], Jelle Bijma[1], Charlotte Cremer[1], Ulf Bickmeyer[1], Zora Mila Colomba Zittier[1]

[1]1Alfred-Wegener-Institut Helmholtz-Zentrum für Polar- und Meeresforschung, Bremerhaven, D-27570, Germany

*Correspondence to*: Jutta E. Wollenburg (jutta.wollenburg@awi.de)

**Abstract.** Deep-sea *Cibicidoides pachyderma* (forma *mundulus*) and related *Cibicidoides* spp. were cultured at in situ pressure for 1-2 days, or 6 weeks to 3 months. During that period, fluorescence analyses following BCECF-AM (2',7'-bis(2-carboxyethyl)-5-(and-6)-carboxyfluorescein acetoxymethyl ester) or Calcein AM (4,5-Bis((N,N-bis(carboxymethy)amino)methyl)fluorescein acetoxymethylester) labelling, revealed a persisting cytoplasmic sheet or envelope surrounding the *Cibicidoides* tests. Thus, the *Cibicidoides* shell can be considered rather as an internal than an external cell structure. A couple of days to a week after being transferred into high-pressure aquaria and adjusted to a pressure of 115 bar, the foraminifera changed from a mobile to a more or less sessile living mode. During this quasi sessile way of life, a series of comparably thick static ectoplasmic structures developed that were not resorbed or remodelled but, except for occasional further growth, remained unchanged throughout the experiments. Three different types of these 'permanent structures' were observed: A) Ectoplasmic 'roots' were common in adult *C. pachyderma, C. lobatulus* and *C. wuellerstorfi* specimens. In our experiments single ectoplasmic 'roots' grew to maximum 700 times the individuals shell diameter and were presumably used to anchor the specimen in an environment with strong currents. B) Ectoplasmic 'trees' describe rigid ectoplasmic structures directed into the aquarium's water body and were used by the foraminifera to climb up and down these ectoplasmic structures. Ectoplasmic 'trees' were so far only observed in *C. pachyderma* and enabled the 'tree'-forming

foraminifera to elevate itself above ground. C) Ectoplasmic 'twigs' were used to guide and hold the more delicate pseudopodial
network when distributed into prevailing currents, and were, in our experiments, also only developed in *C. pachyderma*
specimens. Relocation of a specimen usually required to tear apart and leave behind the rigid ectoplasmic structures, eventually
also the envelope surrounding the test. Apparently, these rigid structures could not be resorbed or reused.
**1 Introduction**
Our knowledge on form and functioning of ectoplasmic extensions in benthic foraminifera is based on laboratory observations
of a few shallow-water species under atmospheric pressure. These studies mostly describe complex networks of branching and
anastomosing pseudopodia that are rapidly and alternately extended and withdrawn into the surrounding environment (Bowser,
2002). The almost continuously remodelling pseudopodia are used for motility, attachment, food collection, the formation of
cysts, growth and certain aspects of reproduction (Goldstein, 1999; Heinz, 2005; Travis, 2002; Tyszka et al., 2019).
Numerous cytoplasmic particles give the pseudopodia a granular appearance when viewed under the light microscope
(Goldstein, 1999). The main components of granule are mitochondria, (secretory, excretory, and storage) vesicles or vacuoles,
and occasionally symbionts (Bowser, 2002). Independently of whether pseudopodia modify their shape or are in a stationary
state, they display constant bidirectional streaming (Bowser, 2002; Rinaldi, 1964). Coupled to this cytoplasmic streaming,
particles are transported bidirectional along the extracellular surfaces of pseudopodia (Bowser, 1985, 1984a). Foraminifera
use this extracellular conveyor belt to collect particles for agglutination or nutrition (Bowser, 2002).
The majority of foraminifera of the genus *Cibicides* (e.g. *C. refulgens, C. antarcticus*) and a significant proportion of
*Cibicidoides* species (e.g. *C. lobatulus, C. wuellerstorfi,* and *C. pachyderma* with the morphotypes *C. pachyderma, C.*
*kullenbergi* and *C. mundulus*, see (Schweizer, 2009) for the genetic versus morphological classification) are epibenthic
(Jorissen et al., 1995; Linke and Lutze, 1993; Lutze, 1989; Nyholm, 1962) although Rose Bengal-stained specimens are
occasionally found at 1-4 cm sediment depth (e.g. (Hunt and Corliss, 1993; Wollenburg and Mackensen, 1998b). However, an
affinity of *Cibicides/-oides* species to settle in places exposed to currents has been inferred from the preferential colonization
of elevated structures exposed to currents or on filter feeding invertebrates (e.g. (Alexander, 1987; Linke and Lutze, 1993;
Schönfeld, 2002). Although facultative grazing on phytodetritus and bacteria on the sediment is proposed for some species
such as *C. antarctica* (Alexander, 1987)  the majority of *Cibicides/-oides* species are assumed to be passive suspension feeders
(Lipps, 1983) trapping phytodetritus by deployment of a pseudopodial network in the prevailing current.
Main target of this study was *C. pachyderma*, of which we continuously observed 57 specimens under *in situ* pressure,
temperature, and current activity conditions over a time span of 3 months. Daily observations allowed us to shed light on the
development of temporary and lasting ectoplasmic extensions in *C. pachyderma*, one of the most important species for palaeo-
reconstructions of the deep sea.
For comparison, 40 *C. lobatulus* and 3 *C. wuellerstorfi* specimens were cultured at corresponding conditions and visually
inspected daily to weekly for a time period of 6 weeks. In addition, fluorescence studies on the ectoplasmic envelope of *C.*
*lobatulus* were carried out for 1-3 days.

## 2 Methods and Material

Central to this study are more or less daily observations on permanent ectoplasmic structures in 57 *C. pachyderma* specimens
that were cultured for 3 months during the 'experiment (1)' of 2017 (Wollenburg et al., 2018). In 2018, we complimented this
data set by daily to weekly observations on permanent ectoplasmic structures in 40 *C. lobatulus* and 3 *C. wuellerstorfi*
specimens cultured for 6 weeks using the same set-up and experimental design as for *C. pachyderma* (Tab. 1). In 2019
fluorescence studies on the ectoplasmic envelope of *C. lobatulus* were carried out for 1-3 days.
High-pressure culturing with small aquaria, like we have used during these experiments, require to keep a stock of foraminifera
at atmospheric pressure for some weeks or months in advance. The decision in favour of *Cibicidoides pachyderma* and *C.*
*lobatulus* species was made as both species live from the shelf to water depths >1000 m and can, thus, be cultured at
atmospheric conditions until they are used in high-pressure experiments. Although it has been shown that barophil *C.*
*wuellerstorfi* is able to survive depressurisation for weeks and can reproduce when subsequently been cultured at *in situ*
pressure (Wollenburg et al., 2015), so far there is no proof that the cell functioning is not altered under such conditions.
During the RV Polarstern expedition PS101 in 2016, pebbles from surface sediments were collected with a multicorer (MUC)
at 79°27.09′N, 7°30.93′E, 856 m water depth and used as stock for the *Cibicidoides pachyderma* experiment (Wollenburg et
al., 2018). During the RV G.O. Sars expedition GS2018108 (Juli -August 2018) pebbles with attached living *C. lobatulus* and





*C. wuellerstorfi* specimens were collected at 900 m water depth on the Norwegian continental slope (68° 00' N, 15° 00' E).
Pebbles of both expeditions were transferred in large lid-covered petri dishes and used as stock cultures for all observations
(see Wollenburg et al., 2018 for handling of the stock cultures). From these stock pebbles, specimens with strong cytoplasm
staining were detached with a cactus-spine under a stereomicroscope, temporarily stored in small (ø 3 cm) seawater-filled petri
dishes in the cold laboratory, and then transferred into the high-pressure aquaria.

| Species | *C. pachyderma* | *C. lobatulus* | *C. wuellerstorfi* |
|---|---|---|---|
| Specimen number | 57 | 40 | 3 |
| Pressure (bar) | 115 ± 1 | 115 ± 1 | 115 ± 1 |
| pH | 8 | 8 | 8 |
| O$_2$ (mmol/L) | 340–396 | 340–396 | 340–396 |
| Tp (°C) | 2.5 ± 0.2 | 2.5 ± 0.2 | 2.5 ± 0.2 |
| Pumping rate (mL/min) | 0.3 (1st month) | 0.3 (week 1-3) | 0.3 (week 1-3) |
|  | 0.6 (month 2-3) | 0.6 (week 4-6) | 0.6 (week 4-6) |
| Feeding (Chlorella/Spirulina) | 0.005 mg weekly | 0.005 mg weekly | 0.005 mg weekly |
| Sediment | partly* | yes | yes |
| Observations | daily | irregular | irregular |


**Table 1.** Basic parameters of the culture experiments. Oxygen and pH values were measured with a combined O$_2$ and pH
measuring device (WTW Multi 3620 IDS) and respective O$_2$ (WTW FDO®925) and pH (SenTix®980) sensors, three times
per week. Fine-grained siliceous oxide (1–5 μm) was used as artificial sediment in one out of four aquaria in the *C. pachyderma*
(*), and in all aquaria of the *C. lobatulus/C. wuellerstorfi* culture experiments.



High-pressure culturing observations on *C. pachyderma* were performed from February to May 2017 (Wollenburg et al., 2018),
observations on *C. lobatulus* and *C. wuellerstorfi* from August to October 2018, and convocal microscope investigations from
October to December 2020.
For this study, a total of 200 L sterile-filtered (0.2 μm mesh) North Sea water was adjusted to a salinity of ~35, by addition of
1 g Hobby Marine sea salt per L and psu-offset, and to a pH of 8.0 under atmospheric pressure. The normal culture seawater
(160 L) was tagged with Calcein AM (4,5-Bis((N,N-bis(carboxymethy)amino)methyl)fluorescein acetoxymethylester) (200
mg/L) to allow for identification of newly precipitated calcite (Wollenburg et al., 2018) and for a better visibility of
foraminiferal protoplasm. To observe ectoplasmic structures under fluorescence light (excitation wavelength of 470 nm,
emission wavelength >490 nm) required to rinse the aquaria with unlabelled seawater from the remaining sterile-filtered batch
of 40 L. This was done every 2–3 weeks for two days. Tagged and non-tagged seawater was stored in multiple 10-L Schott
glass bottles with Bola-connections in a cold room and refrigerator running at 2.5°C. A high-pressure pump (ProStar218
Agilent Technologies) was used to supply a continuous one-way isobaric and isocratic seawater flow through the serially
arranged aquaria running at an experimental pressure of 115 bar. Weekly, with a second high-pressure pump, 0.005 mg of
dried *Chlorella* and *Spirulina* algae dispersed in seawater were pumped in each individual aquarium containing foraminifera
(Wollenburg et al., 2018).
*Cibicidoides* specimens and the development of momentary and durable ectoplasmic extensions were observed under a Zeiss
Axio Zoom V16 microscope and pictures were taken with an Axiocam 506 colour camera.
In 2019, 1 to 3 day-lasting high-pressure (100 bar) fluorescence measurements with *C. lobatulus* were performed. For these
investigations, *C. lobatulus* specimens from the 2018 stock were transferred in a ~10 mL aquarium with windows on both
sides and installed in a portable cooling table running at 1.5°C. A volume of 0.6 mL/min of non-labelled culturing water was
directed through the high-pressure aquarium. For examination, a Confocal- Leica TCS SP5 II equipped with a HCX PL Fluotar
objective (10x/0.30) and an argon laser (λex = 488 nm) was used. Fluorescence emission was measured at 494 - 504 nm. The
assessment and evaluation of the images were done with the software LAS AF Lite (Leica Camera AG). A stock solution of
BCECF-AM (2',7'-bis(2-carboxyethyl)-5-(and-6)-carboxyfluorescein acetoxymethyl ester) in DMSO (1 mg/mL in
dimethylsulfoxid) was mixed and stored at -20 °C. Prior to the staining procedure, control observations were made to check



for foraminiferal autofluorescence. At used microscope settings there was no autofluorescence of *C. lobatulus* specimens prior
to staining. For incubation, the selected specimens were transferred into a petri dish with 2 mL seawater and exposed to 5
µmol/LBCECF-AM. The incubation medium was then gently stirred with a small brush to distribute the dye evenly. The petri
dish was covered and stored at 4 °C for 19 hours (incubation time). The properties of BCECF-AM allow to conduct a non-
terminal life-dead screening procedure (Bernhard et al., 1995). The nonfluorescent membrane permeable BCECF-AM enters
an organism and has to be converted to fluorescent BCECF via intracellular hydrolases, thus, the cell has to be alive to exhibit
fluorescence. After incubation, specimens were transferred into the high-pressure aquaria and gradually adjusted to a pressure
of 100 bar over a period of 6 hours. The observations were conducted right after the aimed pressure was reached, after 24
hours, and after 48 hours. The settings from the control measurement were used to record the fluorescence activity in the
cytoplasm of the *C. lobatulus* specimens. As the *Cibicidoides* test proved to be too thick to be penetrated by the argon laser,
only ectoplasmic features could be investigated with the confocal microscope.

**3 Results**
As the refraction index of foraminiferal cytoplasm approximates that of water, pseudopodia and other cytoplasmic extensions
are usually observed with inversed microscopes once they are in contact to or close to the thin glass bottom of the observational
dishes (e.g. (Bowser, 2002; Cedhagen and Frimanson, 2002; Röttger, 1982; Travis, 2002). High-pressure culturing requires a
thick glass and a certain interior aquarium height, in our case both measuring 4 mm. In these aquaria thin pseudopodia could
only be observed occasionally when a specimen positioned itself or the respective ectoplasmic structure close to the aquarium's
window. Therefore, our results do not comprise a comprehensive documentation of the fine branched parts of the pseudopodial
network but essentially of the thicker ectoplasmic structures.
**3.1 Shell envelope**
At all times, all *Cibicidoides* tests were covered by a thin to thick continuous layer of ectoplasm (envelope) making the shell
an internal rather than an external cellular structure (Figs. 1-2). The shell envelopes showed numerous granules, and in this
respect resembled the appearance of pseudopodia (Fig. 1a-d). Although at an extremely low speed (significantly less than <10



µm per 10 min), the envelope-inherent granules gradually changed their position over time. A coherent ectoplasmic structure
of the shell envelope is corroborated by BCECF-AM staining / confocal microscope analyses (Fig. 1e1-e2). Extension of
pseudopodia from the shell envelope became apparent when algae adhered to these filaments during feeding (Fig. 1c-d),
whereas hours to days after feeding a significant portion of the fed algae were found covering parts of the shell envelope. We
assume that the shell envelope initiates the formation of the agglutinated cyst that covers *Cibicidoides* tests during shell
precipitation/growth or in waters of low pH (De Nooijer et al., 2009; Wollenburg et al., 2018). Similarly, a pure algae-half
cyst formed during a period of 6 weeks on the spiral side of an adult *C. lobatulus* (Fig. 2a-b). Figure 2a shows a bright shell
envelope covering the umbilical side of the specimen and the algae cyst with ectoplasmic contributions on the spiral side. After
6 weeks, the half cyst was shed but still showed parts of what we assume to be ectoplasmic remains (Fig. 2b). Occasionally (n
= 2) also abandoned ectoplasmic envelopes were observed, supporting the idea that the cytoplasmic envelope serves as matrix
for the cyst formation (Figs. 2c-d).







**Figure 1.** Shell envelope I. (a-b) Shell envelope (ee) of a *Cibicidoides pachyderma* specimens revealing multiple granule (g)
and initial static ectoplasmic structures (les). (c-d) Shell envelope of a *C. pachyderma* specimen 24-hours before (c) and during
feeding (d). During feeding multiple mobile granule and attached algae (a) indicate a pseudopodial network presumably
originating in the shell envelope. (e-1-e-2) BCECF-AM incubated *C. lobatulus* specimen viewed under normal transmitted
light (e-1) and laser excitation exhibiting the BCECF-AM fluorescence (e-2). As *C. lobatulus* specimens possess a thick shell,
only the shell envelope, an initial lasting ectoplasmic structure (les), and especially granule reveal bright red fluorescence.





**Figure 2.** Shell envelope II: (a) Shell envelope apparent on the umbilical side of an adult *C. lobatulus* specimen, whereas an algae half-cyst was formed over a period of six weeks over the spiral side. (b) The half cyst 1-2 days after it has been abandoned (cyst was shed during the weekend). (c-d) Abandoned shell envelope of a *C. pachyderma* specimen retrieved after the termination of the *Cibicidoides pachyderma* experiment. (c) and (d) show the same cyst but different focussing. ee= ectoplasmic envelope, e = remains of ectoplasm.



### 3.2 Static ectoplasmic structures

Within 24 hours after transfer into the aquaria and adjustment to a pressure of 115 bar, the first type of thick static ectoplasmic

structures, ectoplasmic 'roots', appeared in about 50% of juvenile and most adult specimens (Figs. 3-9). Juvenile *Cibicidoides*

specimens were more mobile than adults (Wollenburg et al., 2018) and likely therefore, the formation of ectoplasmic 'roots'

was often delayed. Three days and two weeks after transfer, first ectoplasmic 'twigs' and 'trees', respectively, were formed

directing into the water column. All static ectoplasmic structures may have shown continued growth but otherwise changed

little over the 3 months of observation. In one case braided ectoplasmic 'roots'even persisted after the termination of the

experiment when the two involved specimens were rinsed in deionized water and dried (Fig. 5g). We never observed that these

structures were in whole or in part resorbed.

### 3.2.1 Ectoplasmic 'roots'

The most frequent static ectoplasmic structures were 'root-like', extending along the bottom or adhering to the window of the

aquarium (Figs. 3-5). Where the ectoplasmic 'root' came close to the aquarium glass, thereby reducing the distance to the

microscope objective, pseudopodia and bidirectional streaming on the outside of the respective ectoplasmic 'root' could be

observed (Fig. 4). Ectoplasmic 'roots' were attached to the aquarium glass via thickened endings (Figs. 3-4). The typical

ectoplasmic 'root' had a mean thickness of roughly 30 µm and often two 'roots' were twisted to form thicker braid-like

structures (Fig. 5). Presumably limited by the dimension of our aquaria, a maximum root length of roughly 5 mm was observed

(Figs. 4-5). Over the course of the experiments, the number of ectoplasmic 'roots' increased and some showed ongoing growth

(Fig. 4). Figure 5a shows a twisted ectoplasmic 'root' with a total length of 400 µm on the left and a shorter straight 'root' of

approx. 100 µm on the right side of *C. pachyderma* specimen 1 (Sp. 1). Both structures had formed in the course of a night.

During the following day, Sp. 1 flipped over so that the test periphery was facing the aquarium floor, and moved to the filter

ring. There the smaller single ectoplasmic 'root' continued to grow and branch (Fig. 5b-c). Finally, this ectoplasmic 'root' of

Sp. 1 combined with the ectoplasmic 'root' of a neighbouring specimen (Sp. 2) and formed a single braid-like ectoplasmic

'root' (Fig. 5d). For the remaining 2 months, the two individuals moved along this braided 'root' like on rails and positioned



themselves sometimes closer to, sometimes further away from each other. Hereby, specimen 2 remained under the filter ring
for most of the time.

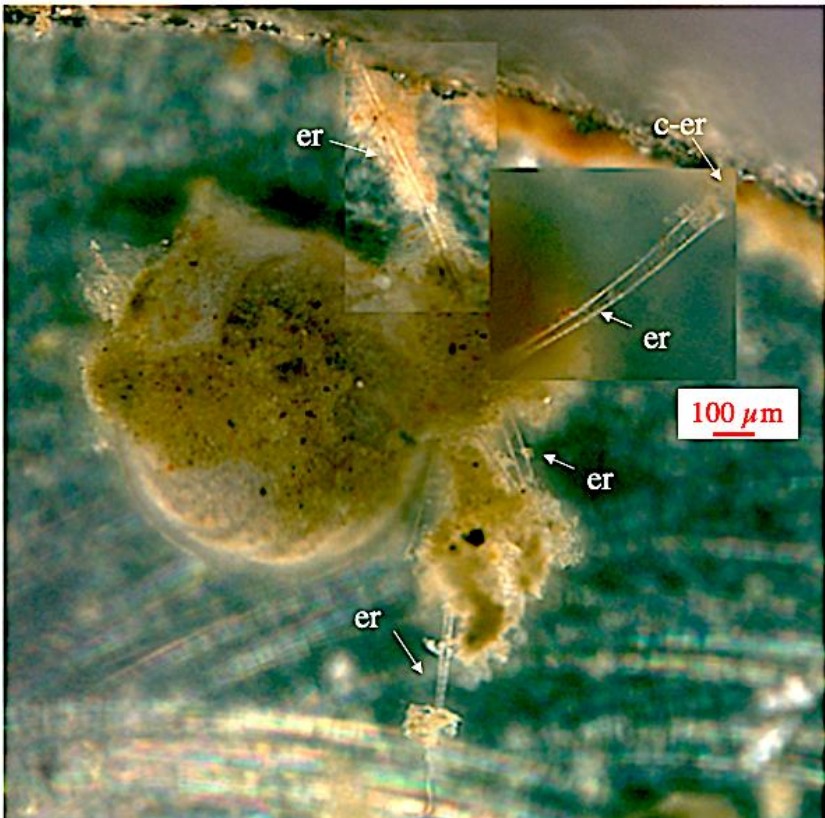


**Figure. 3.** Ectoplasmic 'roots' of *C. wuellerstorfi*. Starting with the lower ones, ectoplasmic 'roots' were developed over a
period of 1 week and remained unchanged for the remaining 5 weeks of the experiment. er= ectoplasmic 'root', c-er= contact
zone of ectoplasmic 'root' with the aquarium glass.




**Figure. 4.** Ectoplasmic 'roots' of *C. lobatulus*. (a) *Cibidoides lobatulus* specimen (sp) embedded in algae with two ectoplasmic

'roots' (er) extending on the bottom of the aquarium. At one point, the northern ectoplasmic 'root' bends upward at the



aquarium's wall, thus, it is differentiated in a lower (l-er) and an upper (u-er) part. (b) Shows the upper part of the northern
ectoplasmic 'root'. (c) Shows the u-er at higher magnification revealing granule (g), pseudopodia (p), and a broad contact zone
(c-er) where the ectoplasmic 'root' is attached to the aquarium's window.








**Figure. 5.** Ectoplasmic 'roots' of *C. pachyderma* (specimens 1 and 2). (a) Six days after being transferred into the high-
pressure aquarium, overnight a twisted ectoplasmic 'root' formed on the left and a short simple 'root' on the right side of the
test of specimen 1 (Sp. 1). (b) Thereafter, Sp. 1 moved towards the filter ring, and finally positioned itself close to an
ectoplasmic 'root' of specimen 2 (Sp. 2; situated under the filter ring) on March 29. (c) The next day, the right-hand ectoplasmic
'root' of specimen 1 started to fray. (d) Several days later, during a weekend, specimen 2 resurfaced from below the filter ring
and its left-hand ectoplasmic 'root' was combined with the frayed right-hand 'root' of specimen 1 to a joined twisted or braided
ectoplasmic 'root'. (e) The joined braided ectoplasmic 'root' of specimens 1 and 2 (positioned under the filter ring) on April
12. (f) Thickness measurements of the joined braided ectoplasmic 'root'. (g) Fluorescence picture of the braided ectoplasmic
'root' of Sp. 1 and 2 immediately after termination of the experiment (excitation wavelength 470 nm, emission wavelength
490 nm). The emitted bright greenish colour of the ectoplasmic 'root' indicates a recent cytoplasmic activity. er= ectoplasmic
'root', f-er= frayed ectoplasmic 'root'.
After termination of the experiment, gently washing the specimens over a 30 μm mesh, and drying the residue, both specimens
were still attached via the joined braided ectoplasmic 'root' with a final length of at least 5 mm (Fig. 5g).

**3.2.2 Ectoplasmic 'trees'**
Thick, robust, and permanent ectoplasmic structures, very similar to ectoplasmic 'roots' but extending into the water column,
were termed ectoplasmic 'trees'. "Tree"-forming *Cibicidoides pachyderma* specimens could climb up these structures to raise
themselves above the bottom. Interestingly, similar structures were not observed in any of the investigated *C. lobatulus* and *C.*
*wuellerstorfi* specimens.
Whereas ectoplasmic 'roots' were eventually formed within 24-hours after transfer into the aquaria, it took about two weeks
before the first ectoplasmic 'trees' were formed (Fig. 6). Rather than moving with the foraminifera, as described for the
'roots'of some specimens, ectoplasmic 'trees' were fixed in the aquaria. They reached a maximum height of approx. 2 mm
and the foraminifera could climb freely along these tree-like structures (Fig. 6a-c). Regularly spaced short and obviously
adhesive side-branches (Fig. 6a), probably with tiny pseudopodia (that are rarely visible in our set-up), collected suspended



algae from the inflow current during feeding. As result ectoplasmic 'trees' looked like loosely agglutinated structures, later in
the experiment (Figs. 6c-1-2).








**Figure 6.** Ectoplasmic 'trees' of *C. pachyderma*. (a) Ectoplasmic 'tree' of *C. pachyderma* specimen (sp.) 3 with three thick branches originating from a single "stem" fixed to the aquarium wall. *Cibicidoides pachyderma* sp. 3 was positioned approx. 100 µm away from the wall with no contact to the bottom of the aquarium. (b-1-3) Ectoplasmic 'tree' of *C. pachyderma* sp. 4 fixed to the aquarium's bottom and extending at least 2 mm into the water column. (b-1) On April 10, specimen 4 had climbed to the top of the ectoplasmic 'tree'. (b-2) The next day, the specimen had moved to the middle section of the ectoplasmic 'tree'. (b-3) Shows, as an example, specimen 4 at the bottom of the ectoplasmic 'tree' on May 28. Furthermore, thickness measurements on the 'tree' structures are provided. (c-1-2) Ectoplasmic 'tree' of *C. pachyderma* sp. 5. Algae adhering to the adhesive side branches of the ectoplasmic 'tree' obscure the ectoplasmic nature when viewed under normal light (c-1). (c-2) Shows the same ectoplasmic 'tree' under fluorescent light, allowing a better visibility of the 'tree' and the specimen's position. The bright greenish fluorescence of the Calcein-labelled cytoplasm illustrates the elevated position of specimen 5 within the accumulated algae.

et= ectoplasmic 'tree', sb= side branches.

### 3.2.3 Ectoplasmic 'twigs' and pseudopodial network

Thick ectoplasmic structures extending into the water were termed ectoplasmic 'twigs' if the shape and position with respect to the test remained essentially permanent during the experiment (Figs. 7-8). However, ectoplasmic 'twigs' are the least static of the three described ectoplasmic structures and were only observed in *C. pachyderma* specimens so far. The first ectoplasmic 'twigs' appeared 3 days after transfer of *C. pachyderma* specimens into the aquaria (Fig. 7a). Additional structures were eventually added over time (Fig. 7a-b), but the original structure was usually not modified (Figs. 7-8). Provided with the same short and obviously adhesive side branches as ectoplasmic 'trees' (Fig. 6), the ectoplasmic 'twigs' probably support a more delicate pseudopodial network (Figs. 7-8). In our experiment, *C. pachyderma* specimens exhibited a strong rheotaxis. In this context it was observed that a specimen had positioned itself at the hole of the filter ring (where the food entered the aquarium). After this position was occupied the specimen developed a series of crescent-shaped ectoplasmic 'twigs' (Fig. 8). From the area in which the ectoplasmic 'twigs' were developed, the species directed an anastomosing pseudopodial network into the





inflowing water current during feeding (Figs. 8-10). In doing so, the instrumentally visible collection area increased by at least
twenty times the specimen's test size. Hereby, both the pseudopodial network and the respective supportive ectoplasmic 'twigs'
obviously allowed the animal to collect food from the water current (Figs. 8-10). When we shut down the pumps and, thus,
the current activity for some minutes (on May 26, 2017, 25 hours after feeding), the pseudopodial network, visualized by
adhering algae, collapsed (Fig. 8b), whereas the ectoplasmic 'twigs' kept their original shape (Fig. 8). The shape of the
specimen's ectoplasmic 'twigs' was neither affected by the presence or absence of the current nor by the speed of it (~0.1-5
cm/min (Wollenburg et al., 2018)).
For the specimen positioned at the hole in the filter ring, the development and extension of pseudopodia directing into the
water current during feeding was immediate (Fig. 10), however, the transport of collected algae towards the shell was extremely
slow. Seven hours after feeding, algae were still sticking to the pseudopodia and ectoplasmic 'twigs' and no or only low
amounts of fresh algae had reached the shell interior (Fig. 10f). Slow food ingestion was also reflected by the extremely slow
propagation of anastomoses over time. An anastomosis propagated less than 150 µm within 24 hours (Fig. 10). During and
following feeding, the number of granules in the ectoplasmic envelope, the ectoplasmic 'twigs', and pseudopodia were
significantly increased.



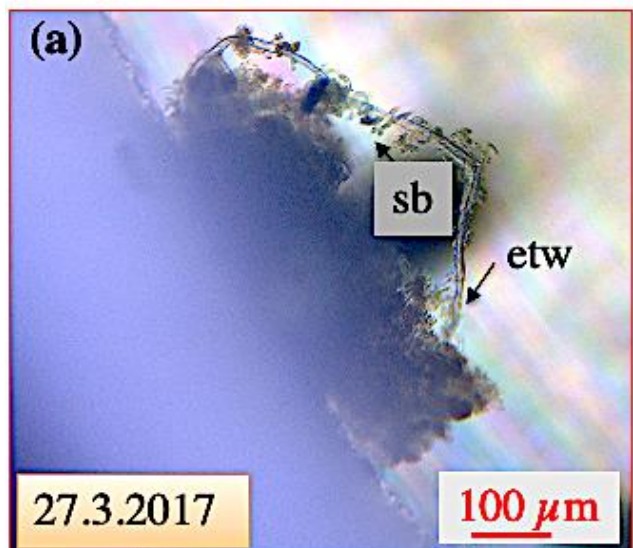

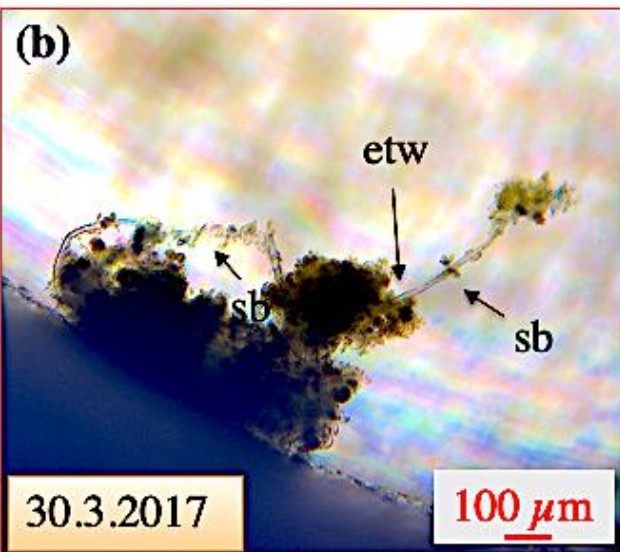



**Figure. 7.** Ectoplasmic 'twigs' of *C. pachyderma* specimen 8. (a) For 3 days, the specimen had gathered algal detritus around
its shell envelope and simultaneously developed a loop-like ectoplasmic 'twig' with a total length of ~700 µm from the
periphery to the opposite side. (b) Three days later, an ~500 µm-measuring extension directing into the water column was
added to the loop-like 'twig'. Both structures persisted for the remaining weeks of the experiment. (c) On May 15, dispersion
of algae into the aquarium allowed the specimen to collect additional algae onto the ectoplasmic 'twig'. The algae mass
remained in this position and was not ingested during the experiment. etw= ectoplasmic 'twig', a= algae, sb= side branch.


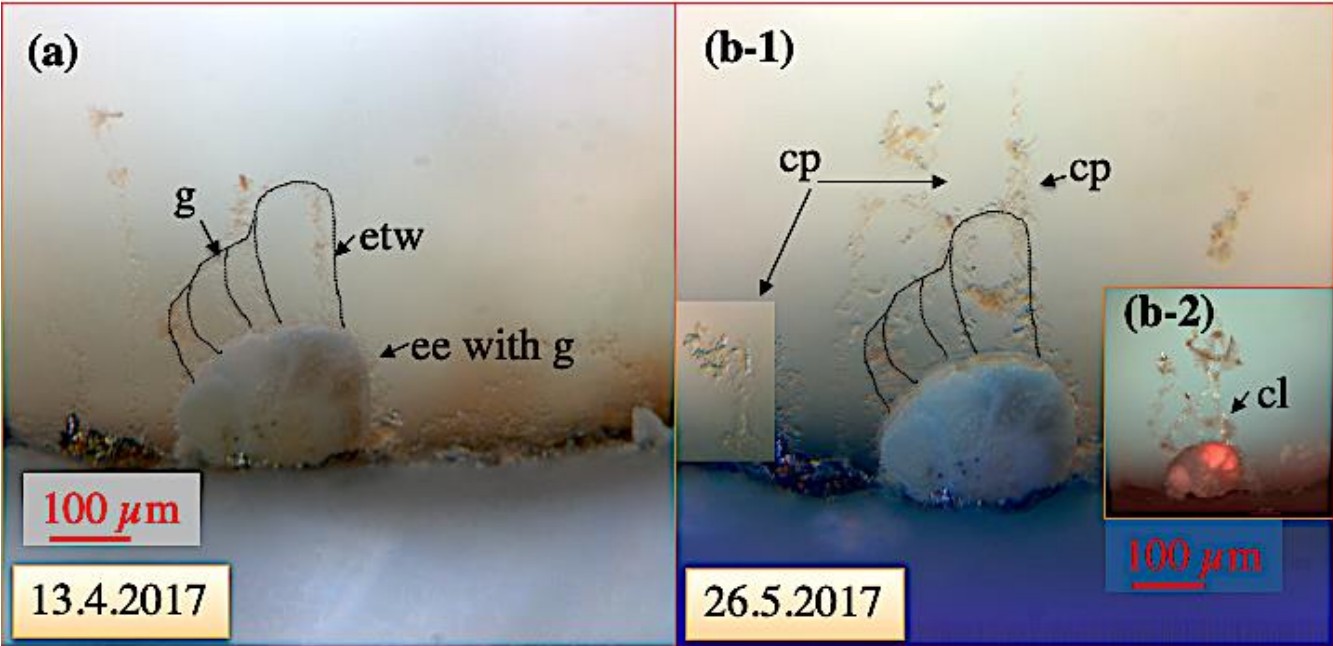


**Figure. 8.** Crescent-shaped ectoplasmic 'twigs' of *C. pachyderma* specimen 1 positioned at the hole of the sinter ring, i.e. at
the inflow of water and algal food into this aquarium. (a) Specimen viewed under normal light when no food was added to the
inflow revealing bow-like ectoplasmic 'twigs'. (b) 35 days later, the pumps were stopped to investigate the stability of the
ectoplasmic 'twigs' and the pseudopodial network at zero current activity. Stable ectoplasmic 'twigs' and collapsed
pseudopodial (cp) network under normal light (b-1) and fluorescent light (b-2). The red colour of especially older test parts



result from ingested *Spirulina* and *Chlorella* algae stored in food vacuoles of the cytoplasm. ee = ectoplasmic envelope, etw=
ectoplasmic 'twig', g= granule, cl= Calcein-stained cytoplasmic lacuna in the etw and cp.







**Figure. 9.** Pseudopodial network of *C. pachyderma* specimen 1 during feeding on April 13 2017.
Specimen 1 before, during, and after feeding with 0.5 µg dried *Spirulina* and *Chlorella* algae. The bright red colour of dispersed
algae under fluorescent light provides an excellent tool to document the passage and uptake of algae in the pseudopodia and
cytoplasm. (a-b) Specimen 1 prior feeding. (c) Schematic illustration of the aquaria indicating the start of feeding. (d) Specimen
1 during feeding. (e-f) Seven hours after feeding. etw= ectoplasmic 'twig', p= pseudopod, a= algae, nl = normal light, fl =
fluorescence light. Numbers state the respective time on April 13.









**Figure. 10.** Pseudopodial network of *C. pachyderma* specimen 1 under fluorescent light on May 25 and 26. Movement of an
anastomosis within 24 hours after feeding. (a-1) In course of the experimental running time, a visually increasing amount of
algae (intensified red colour of cytoplasm; compare to Fig. 9) had accumulated in the specimen's cytoplasm. A red square
indicates the position of a slowly moving anastomosis in the pseudopodial network. (a-2) Shows the test at higher magnification
revealing the presence of numerous granules in the ectoplasmic envelope and 'twigs'. (b) 24 hours later, the anastomosis had
moved by approximately 150 µm towards the shell. an= anastomosis, ee = ectoplasmic envelope, g= granule.

**3.2.4 Torn ectoplasmic remains**
When *Cibicidoides* specimens that were virtually sessile for weeks changed position, their static ectoplasmic structures could
obviously not be resorbed. These structures were either pulled along by the specimens, as shown for the ectoplasmic 'roots' in
Fig. 5, or torn off. Over the duration of the experiment, numerous ectoplasmic 'roots' and 'twigs', or what is supposed to be
parts of such structures, were flushed to the aquarium's window (Fig. 11). We had to increase the current speed through the
aquaria sporadically to get rid of the torn biomass and clear the view.

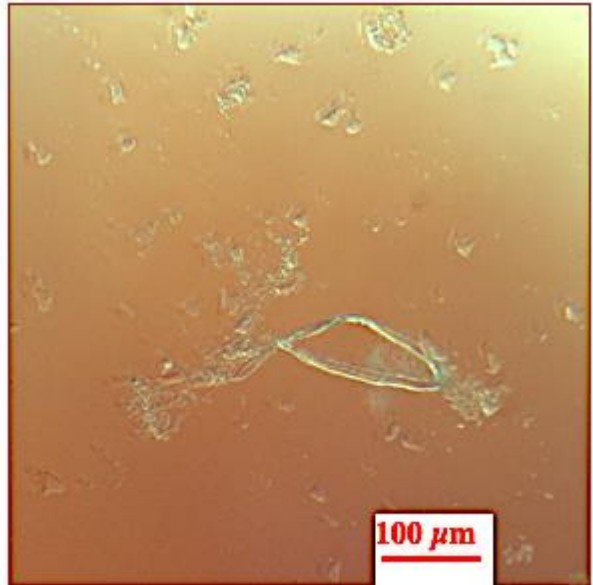


**Figure. 11.** Torn ectoplasmic 'roots' and 'twigs' at the aquarium window on May 2, 2017.



## 4 Discussion

This study is the first to describe the shell of *Cibicidoides* spp. as an internal 'sceleton' rather than an external feature. However, the observation of an ectoplasmic sheet or envelope around foraminiferal shells goes back to the early days of foraminiferal observations when it has been described for *Heterostegina depressa* (Röttger, 1973, 1982). The observation of a significantly reduced pH surrounding *Ammonia* sp. shells during growth (Toyofuku et al., 2017) may point to an envelope also in *Ammonia*. However, so far, no sheet or envelope has been described for this most studied genus. There are also some vague parallels between ectoplasmic envelopes and the Actin-rich lamellipodia that cover the tests of *Amphistegina lessonii* specimen during chamber formation (Tyszka et al., 2019). Yet, in our observations, an ectoplasmic envelope covered the tests of the investigated *Cibicidoides* specimens at all times and for shell growth a supplementary surrounding sediment cyst had to develop (Wollenburg et al., 2018). Thus, it is currently unclear whether an ectoplasmic envelope is developed in only a few foraminifera taxa or has simply been overlooked in others. As described for *Hetereostegina depressa* (Röttger, 1982), also in our experiments the *Cibicidoides* specimens obviously only shed their envelope during rapid relocation.

Only for a few shallow-water benthic foraminifera, information on ectoplasmic extensions to interact with the environment has been published so far (Bowser, 2002; Travis, 2002). Hereby, the typical ectoplasmic extensions described are pseudopodia characterised by their forceful and rapid extension enabled by actin filaments and extremely dynamic microtubule systems (Bowser et al., 1988; Goleń et al., 2020; Travis and Bowser, 1986; Travis, 2002). Anastomosing, i.e. the fusing of two neighbouring pseudopodia, is abundant and rapidly propagating. Furthermore, a rapid bidirectional transport of both granules and surface-attached particles has been described for the pseudopodia of shallow-water foraminifera. Giving tribute to the granular appearance, the term 'granuloreticulopodia' is widely used for this pseudopodial network and separates it from the globular and lamellar pseudopodia involved in chamber formation (Goleń et al., 2020; Tyszka et al., 2019).

Our study shows that at in situ pressure the pseudopodial network of the examined *Cibicidoides* taxa extends into the water current and exhibits branching and anastomoses, resembling the pseudopodial network of shallow-water foraminifera. However, in the investigated specimens granules, anastomoses, and attached particles moved very slow and could be observed for hours, sometimes even days or weeks with little noticeable movement (Figs. 9-10). In *C. pachyderma* sp. 1 of Figs. 8-10,

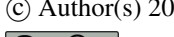



for example, it took about 6 weeks before a significant ingestion of dispersed algae inside the shell could be noticed (Figs. 9-

330  10).

The rate at which cells can form projections, like pseudopodia, and transport granules and adhering particles is, in part, limited
by the rate at which the cell assembles new or reorganises existing actin filaments (Bowser et al., 1988; Goleń et al., 2020;
Travis and Bowser, 1986; Travis, 2002; Tyszka et al., 2019). This ATP consuming process is obviously much faster in shallow-
water foraminifera than in deep-water *Cibicides/Cibicidoides*-taxa. Presumably due to the large working distance in our high-
pressure aquarium set-up fluorescent SiR-actin labelling failed in our confocal studies so far. Therefore, we can just speculate
that the ATP demand to form pseudopodia and perform bidirectional streaming increases with hydrostatic pressure and/or at
sites of high current activity.
Besides pseudopodia, this study describes for the first time non-retractable static ectoplasmic structures that, depending on
their characteristics, were named ectoplasmic 'roots', 'trees', and 'twigs'. Ectoplasmic 'roots' developed in most specimen
and all species investigated. Hereby, minimum 2 mutually opposing ectoplasmic 'roots' developed soon after the start of the
experiments. However, over the course of the experiments, the number of ectoplasmic 'roots' increased and most showed
ongoing growth. Ectoplasmic 'roots' are long branchless structures extending along the bottom or adhering to the window of
the aquarium. Together with pseudopodia emerging from the ectoplasmic 'root', these structures likely act as anchors to
stabilize the foraminiferal shell in an area of high current activity. Ectoplasmic 'roots' are likely the 'naked' variant of the
agglutinated tubes of *C. lobatulus* described from shallow-water occurrences (Nyholm, 1962). We assume that similar to the
sedimentary cyst covering the ectoplasmic envelope (see above), deposition of current-collected sediment particles on top of
ectoplasmic 'roots' leads to an increased robustness and protection of these structures.
Ectoplasmic 'trees' are thick, robust, and branching structures that, other than 'roots', direct into the water column (Fig. 6).
Over the course of weeks in the experiments, ectoplasmic 'trees' were only formed by *C. pachyderma* specimens. Fixed to the
aquarium bottom, these protruding structures reached heights of around 2 mm. Ectoplasmic 'trees' likely serve as scaffolding
on which the foraminifera can modify or optimise its position with respect to the prevailing current.
Ectoplasmic 'twigs' are thick structures extending into the water column whose shape and position with respect to the
specimen's test remain largely unchanged. However, they are the least static ones of the three described ectoplasmic structures.





Ectoplasmic 'twigs' are perhaps a stabilizing and protective framework that maintains a delicate pseudopodial network when
distributed into a current. However, further studies are required to prove our assumptions. In our high-pressure experiments,
ectoplasmic 'twigs' were only observed in *C. pachyderma* specimens, yet, recent observations on shallow-water *C. lobatulus*
show 'agglutinated' tubes directing into the water column (Fig. 12) that resemble ectoplasmic 'twigs'. In Fig. 12 we see a joint
'agglutinated' tube between specimen 1 (juv. *C. lobatulus*) and 2 (adult *C. lobatulus*) with freshly (picture was taken following
a feeding experiment) accumulated algae half way. On specimen 2 a second 'agglutinated' tube directs into the water column.
From our experience with cyst formation and algae aggregation, we assume that these 'agglutinated' tubes are sediment
covered ectoplasmic 'twigs'. If *C. lobatulus* just develops ectoplasmic 'twigs' at shallow-water/ low-pressure sites, or if they
were too thin to be detected with our instrumental set-up in our experiments with this species remains unclear. However, the
picture of these freshly fed shallow water *C. lobatulus* specimens supports our assumption that the formation of rigid
ectoplasmic 'twigs' assists a food-gathering pseudopodial network.

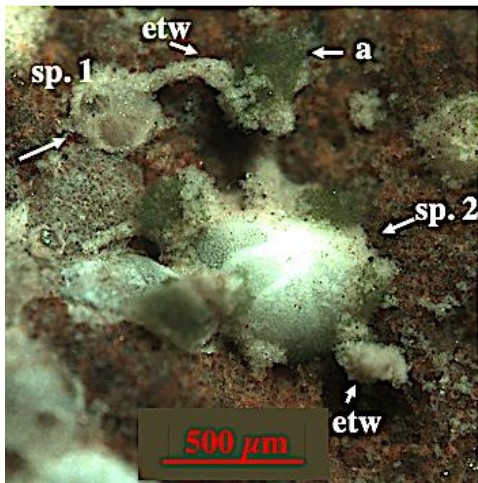


**Figure. 12.** Epilithic *C. lobatulus* specimen from off Svalbard. A joined 'agglutinated' tube, here equated with ectoplasmic
'twigs', is developed between specimen 1 and 2. Algae are accumulated half-way the tube. etw= ectoplasmic 'twig', a= algae.
Picture courtesy of Julia Wukovits (September 2020).

We observed that static ectoplasmic structures did not change in response to current speed and that they could not be resorbed
or retracted. When we opened the aquaria after termination of the experiments, we found torn ectoplasmic 'roots' with no signs



of shrinking or collapsing. Since static ectoplasmic structures can obviously not be resorbed, any relocation is accompanied

by material loss for a specimen.

It was also observed that algae (dispersed from the water inflow) adhering to the static ectoplasmic envelope, 'twigs', 'trees',

and less marked 'roots', remained almost at the same position throughout the experiment or until the respective structure was

torn off (Figs. 6-7). This might suggest that, in the absence of sediment particles in the current, the foraminifera try to stabilise

lasting ectoplasmic structures by the continuous accumulation of algae (see also below).

In the field, the pseudopodial network of *C. antarcticus* is assumed to be guided by agglutinated tubes extending from the

foraminiferal shell into the water column (Alexander and DeLaca, 1987b; Alexander, 1987; Hancock et al., 2015). In our

experiments the ectoplasmic 'trees' and 'twigs' accumulated algae over time, but likely would also have accumulated

sediments if provided by the inflowing current. Hypothetically, accumulation of sediment particles on ectoplasmic 'twigs' and

'trees' over longer periods could result in structures that resemble the agglutinated tubes described for *C. antarcticus*

(Alexander and DeLaca, 1987b) or shallow-water *C. lobatulus* (Fig. 12).

The tubes of *C. antarcticus* are made up of silt- and clay-sized minerals, diatom frustules, fine organic detritus, and occasionally

sponge spicules. However, although being described as agglutinated structures, the tubes collapsed when the respective

foraminifera was taken out of the water (Alexander and Delaca, 1987a). As no analyses on the particle combining cement were

made, it is quite possible that the described agglutinated tubes are sediment-covered ectoplasmic structures. In our study

provided artificial quartz substrate was not used for agglutination or accumulation on the static ectoplasmic 'roots, 'trees', or

'twigs', whereas dispersed algae were collected from the inflowing current and deposited on these structures. As we had no

dispersed minerals in the circulating current it can only be assumed that they would also adhere to the lasting ectoplasmic

structures described.

Bowser and Travis (2002) speculated that evolutionarily the pseudopodium may have derived from the eukaryotic flagellum

because nearly all foraminifera possess flagellated gametes (Goldstein, 1999). Both, flagella and pseudopodia rely on

microtubules as a supporting and locomotive framework. Flagella possess an elaborate crosslinking apparatus designed to

produce a highly regulated bending form, whereas in shallow-water foraminifera microtubules are constantly transported

within the tethered framework of pseudopodia allowing a less rigid but highly flexible motile function. Although, pseudopodia



emerged from the static ectoplasmic structures, due to the stiffness of 'roots', 'trees', and 'twigs', they rather resemble flagella
than pseudopodia. Yet, future transmission electron analyses or confocal microscope investigations at atmospheric pressure
(Goleń et al., 2020; Tyszka et al., 2019) are needed to understand the cellular structure of these lasting ectoplasmic extensions.
Application of fluorescent dyes for confocal microscope investigations in high-pressure aquaria is often limited by the large
working distance hampering e.g. a noticeable emission from SiR-actin labelling.
The static ectoplasmic features described are long-lasting and, thus, presumably energy saving structures of taxa living under
significant hydrostatic pressure and current activity. They likely anchor the specimen at low energetic costs in a highly
turbulent environment. Furthermore, 'twigs' and 'trees' likely protect a delicate pseudopodial network that, in a habitat with
unpredictable food supply has to be immediately developed and extended. However, movement of anastomoses, adhering
algae, and bidirectional streaming in the pseudopodial network were extremely slow during our observations suggesting a
much slower ingestion time than has been described for shallow-water foraminifera (Bowser, 1984a, 2002; Wollenburg et al.,
2018). This may be the reason why, for example, *C. wuellerstorfi* in the Nordic Seas and Arctic Ocean is restricted to times
and areas of high food supply but is insensible to sudden primary production/carbon export pulses (Wollenburg and Kuhnt,
2000; Wollenburg et al., 2001; Wollenburg and Mackensen, 1998a).

**5. Summary**
This is the first report investigating ectoplasmic structures and dynamics in *Cibicidoides* species under *in situ* pressure. In the
present study, a protective ectoplasmic envelope completely covered all *Cibicidoides* shells at any time suggesting that the
shell is an endo- rather than ectoplasmatic feature.
Our further findings indicate that the life of these deep-sea foraminifera is characterised by energy-saving, long-lasting, static
ectoplasmic structures that allow these rheotactic species to position themselves at sites of high current activities. 'Roots' are
thick and robust ectoplasmic structures that anchor the specimens on current exposed substrates. They might continue to grow
but otherwise could not be reshaped. Ectoplasmic 'trees' are stationary structures that are directed into the water column
allowing the foraminifera to climb this structure and thereby elevate itself above ground.



Ectoplasmatic 'twigs' provide a supportive rigid framework from which or around which a delicate food-gathering
pseudopodial network emerge.
When the specimen changed their location, the stationary ectoplasmic 'trees' and one or the other ectoplasmic 'root' were torn
off. Thus, relocation is associated with a loss of ectoplasm and an additional energy demand required for the formation of new
lasting ectoplasmic structures to secure the specimen at its new location. Whereas the deployment of a pseudopodial network
into an inflowing current with algae is immediate, the propagation of collected algae towards the shell is extremely slow.
Perhaps for this reason *Cibicidoides* taxa are poor indicators of primary production pulses.
We assume that the static shape and slow remodelling of 'trees', 'twigs', and 'roots' as well as the slow formation of
anastomoses and surface transport arises from an adaptation to a high current activity habitat with unpredictable food fluxes
driven by energetic optimization. This assumption as well as the possibility of a different microtubule system in deep-sea
pseudopodia have to be addressed in future studies.

*Acknowledgments*
We thank the chief scientist Prof. Antje Boetius and the captain and crew of RV Polarstern for their fantastic work during cruise
PS101. Special thanks go to Linn Schmidtman who did a wonderful job in collecting the foraminiferal material during this
expedition. A big 'thank you' goes to Erik Wurz who collected stones with foraminifera during the RV G.O. Sars expedition
GS2018108 (Juli -August 2018) for this study. During her stay at the AWI Julia Wukovits took the wonderful picture of living
*C. lobatulus* that she allowed us to use in this publication. We are extremely grateful for the excellent work of the AWI
workshop, and especially to Erich Dunker who was always at our side designing and redesigning the high-pressure aquaria and
most of the corresponding equipment. In 2018 Johannes Lemburg replaced Erich Dunker and he designed the wonderful
convocal high-pressure aquaria and supportive systems, many thanks!
This study was funded by a DFG-grant to JW (WO742-2), a joined DFG-ANR grant to JB (Project "B2SeaCarb"; BI
432/10-1) and the "CoPter" project (AWI Strategiefond) and through the SponGES-Deep-sea Sponge Grounds Ecosystems
of the North Atlantic: an integrated approach towards their preservation and sustainable exploitation, under H2020 - the EU
Framework Programme for Research and Innovation (Grant Agreement no. 679849).



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
