# Peer review of "Permanent ectoplasmic structures in deep-sea *Cibicides/oides* taxa – long-term observations at in situ pressure"

_Biogeosciences, 2021_

## Author Response (AR1)

Response to
Comment of Susan T. Goldstein and an anonymous (Referee) **(response in bold)**

**We are very grateful for the very positive and helpful comments of Sue Goldstein and an anonymous that helped to improve the manuscript. We addressed all comments in the revised version.**

The authors report observations on living specimens of several species of *Cibicides-/oides*, species that are important epibenthic taxa widely used in paleoceanographic studies. Individuals were maintained at ambient or near ambient pressures in a highly specialized culturing system with circulating seawater. The authors report highly resistant structures formed by the individual foraminifera, and the ability of individuals to move along some of these structures. Although observations are limited by the thickness of the walls of the culture system, they are nonetheless novel and demonstrate the extent to which these foraminifera modify their surroundings by constructing a useful "scaffolding" to aid in motility and feeding.

How many of the individuals observed made these structures?
**We added the respective information to the respective positions in the results chapter. To 3.2, we added 'In 68 out of 100 specimens ectoplasmic 'roots' were observed. In an unknown proportion of the rest (32 specimens), such structures might have existed but due to the large working distance and/or a less optimal observational position of the specimens in the aquarium not noticed.' To. 3.2.2., we added 'Distinct ectoplasmic 'trees' were observed in 6 of the 50 studied *C. pachyderma* specimens, others might have been overlooked as the experimental set-up just allows a vertical view insight the aquarium.' To 3.2.3, we added 'Ectoplasmic 'twigs' are directed above the umbilical side into the water column, thus, in our experiments they could only be observed in specimens that had attached themselves on an, in respect to the observation, ideal position on the aquarium's wall. In 16 of the 50 observed *C. pachyderma* specimens ectoplasmic 'twigs' were observed. '**

It would be good to know what these structures are composed of and whether these materials might be unique to these particular species. Perhaps this will be addressed in future research.
**Yes, it is impossible to perform fixations for transmission electron microscope analyses in our high-pressure aquaria. Such investigations have to be performed in different settings at atmospheric pressure.**

I have just a few comments:

1. Who was the first to describe the granular appearance of reticulopodia in foraminifera? The authors cite Goldstein (1999), but this has been known for quite a long time. I (vaguely) recall reading this description in Rhumbler (1909). Given the content of the paper, I suggest that the authors track down the origin of this observation.

**According to our research the first notification was Schultze (1854). We added this reference and Hedley (1964) as additional literature examples. We also added Dujardin's (1835) work which led to the term Rhizopoda.**

2.  Who was the first to refer to the foraminiferal shell as a test that is internal? Cushman (1948) talks about the test as an internal structure, and judging from the associated illustration, the observation may date as far back as Schultze (1854). Again, if this point is to be reviewed in the Discussion, the authors should track down these earlier descriptions.

**From what we could figure out from the old papers, Schultze (1854) illustrated an 'ectoplasmic sheet' around *E. macellum*, but did not any further comment on that. On the other hand, Cushman (1929, older version of the 1948er edition) stated that in many taxa the foramiferal shell would an internal one but did not elaborate on details and affected species. We added this information, and that in more recent years it is regarded as a structure limited to a certain area of the test and to the time of new chamber formation. The respected text part at the beginning of the discussion now reads 'This study describes the shell of *Cibicidoides* spp., as an internal 'sceleton' rather than an external feature. Already in Schultze's work from 1854 an ectoplasmic sheet can be suspected to cover the illustrated *Elphidium macellum* (as *Polystomella strigilatum*) test (Schultze, 1854)plate IV, fig. 1). Cushman (Cushman, 1928) even stated that in many taxa the foramiferal shell would an internal one but did not elaborate more on which species he had in mind. In studies on foraminiferal calcification processes, in planktonic foraminifera, *Ammonia* sp., and *Amphistegina lessoni* a protective  cytoplasmic envelope is described as a structure restricted to times and areas when/where new shell material is precipitated (Bé et al., 1979; de Nooijer et al., 2014; Erez, 2003; Tyszka et al., 2019).**

3.  When the pumps are 'shut down', does the pressure also drop, or does this just affect the currents?

**A shut-off valve is installed after the overflow valve to ensure that pressure doesn't drop once pumps are shut-off. We added a respective footnote to the legend of Fig. 8 ('A shut-off valve following downstream the overflow valve prohibited a pressure drop in the high-pressure aquaria when the pumps were shut.')**

4. A couple of typos: line 83, convocal: confocal; line 121: inversed: inverted.

**We corrected the typos accordingly.**

5.  The authors used calcein, but I didn't see any mention of the occurrence of calcification. Did any calcification occur?

**No, as stated in Wollenburg et al. (2018b) growth of *Cibicidoides mundulus* required the presence of artificial sediments. The observations in this study were done on aquaria without artificial sediments.**

6.  "Et al." has been omitted from a number of the in-text citations, and these should be corrected.

**We corrected the author lists in endnote so that now 'et al.' is displayed.**

Comment of Anonymous Referee #1 (response in bold)

The authors provide new observations on ectoplasmic structures in deep-sea foraminifera and it is the first to describe the shell of Cibicidoides as internal rather than an external feature. They further describe how these structures are used as scaffolding for activities such as motility and feeding.

There is a lot less known on the deep-sea species compared to other groups of foraminifera which makes any new observations an important contribution, and specifically when culturing is done under in situ pressure. Thus, I think these observations are important for further understanding of the physiology and ecology of deep sea benthic foraminifera and only have a few suggestions that might help with clarity

Introduction: at the end of section (line 52-54) it's unclear to me why they did these extra experiments. The authors should consider rephrasing to include the aim

**We wanted to unravel whether the observed features were unique to *C. pachyderma* or common features of the genera *Cibicides/-oides*. BCECF-AM labelling just works for living cell parts, thus, we conducted confocal microscopy with this label to ensure that the specimens were completely surrounded by living cytoplasm not dead tissue.**
**We have modified the paragraph and it now reads 'To determine if the observed ectoplasmic structures are unique to *C. pachyderma* or common to the related genera *Cibicides* and *Cibicidoides*, 40 *C. lobatulus* and 3 *C. wuellerstorfi* specimens were cultured at corresponding conditions and visually inspected daily to weekly for a time period of 6 weeks. To prove that shells were covered by living cytoplasm, in addition, fluorescence studies on the ectoplasmic envelope of *C. lobatulus* were carried out for 1-3 days.**

Method: This section starts with a statement that central to this study are observations from a previous study but don't mention what these are. If they are central, maybe they should have been introduced before, perhaps even in the introduction part.

**This obviously is a misunderstanding, as this manuscript is the first to describe the ectoplasmic structures. As the cited paper addressed different aspects of this experiment, we deleted the reference here to prohibit any confusion.**

Results: The observation are described in much details and combined with the images report clearly the development of the ectoplasmic extensions. However it is not mentions if the observations were done on all specimens and if not on what proportion of them.

**We added the respective information to the respective positions in the results chapter. To 3.2, we added 'In 68 out of 100 specimens ectoplasmic 'roots' were observed. In an unknown proportion of the rest (32 specimens), such structures might have existed but due to the large working distance and/or a less optimal observational position of the specimens in the aquarium not noticed.' To. 3.2.2., we added 'Distinct ectoplasmic 'trees' were observed in 6 of the 50 studied *C. pachyderma* specimens, others might have been overlooked as the experimental set-up just allows a vertical view insight the aquarium.' To 3.2.3, we added 'Ectoplasmic 'twigs' are directed above the umbilical side into the water column, thus, in our experiments they could only be observed in specimens that had attached themselves on an, in respect to the observation, ideal position on the aquarium's wall. In 16 of the 50 observed *C. pachyderma* specimens ectoplasmic 'twigs' were observed. '**

Discussion: This section was a bit hard to follow, will the authors consider dividing it to sub sections with headings? this will help the reader follow each part.

**As requested, we have divided the discussion in subsections.**

Some parts of the discussion might be better suited in the results parts (for example

lines 371-376)

**We followed the suggestion and moved these sentences to the results.**